# Genetic Manipulation of *Desulfovibrio ferrophilus* and Evaluation of Fe(III) Oxide Reduction Mechanisms

Toshiyuki Ueki,[a]* Trevor L. Woodard,[a] (iD) Derek R. Lovley[a,b]

[a]Department of Microbiology, University of Massachusetts-Amherst, Amherst, Massachusetts, USA
[b]Institute for Applied Life Sciences, University of Massachusetts-Amherst, Amherst, Massachusetts, USA

**ABSTRACT** The sulfate-reducing microbe *Desulfovibrio ferrophilus* is of interest due to its relatively rare ability to also grow with Fe(III) oxide as an electron acceptor and its rapid corrosion of metallic iron. Previous studies have suggested multiple agents for *D. ferrophilus* extracellular electron exchange including a soluble electron shuttle, electrically conductive pili, and outer surface multiheme *c*-type cytochromes. However, the previous lack of a strategy for genetic manipulation of *D. ferrophilus* limited mechanistic investigations. We developed an electroporation-mediated transformation method that enabled replacement of *D. ferrophilus* genes of interest with an antibiotic resistance gene via double-crossover homologous recombination. Genes were identified that are essential for flagellum-based motility and the expression of the two types of *D. ferrophilus* pili. Disrupting flagellum-based motility or expression of either of the two pili did not inhibit Fe(III) oxide reduction, nor did deleting genes for multiheme *c*-type cytochromes predicted to be associated with the outer membrane. Although redundancies in cytochrome or pilus function might explain some of these phenotypes, overall, the results are consistent with *D. ferrophilus* primarily reducing Fe(III) oxide via an electron shuttle. The finding that *D. ferrophilus* is genetically tractable not only will aid in elucidating further details of its mechanisms for Fe(III) oxide reduction but also provides a new experimental approach for developing a better understanding of some of its other unique features, such as the ability to corrode metallic iron at high rates and accept electrons from negatively poised electrodes.

**IMPORTANCE** *Desulfovibrio ferrophilus* is an important pure culture model for Fe(III) oxide reduction and the corrosion of iron-containing metals in anaerobic marine environments. This study demonstrates that *D. ferrophilus* is genetically tractable, an important advance for elucidating the mechanisms by which it interacts with extracellular electron acceptors and donors. The results demonstrate that there is not one specific outer surface multiheme *D. ferrophilus* *c*-type cytochrome that is essential for Fe(III) oxide reduction. This finding, coupled with the lack of apparent porin-cytochrome conduits encoded in the *D. ferrophilus* genome and the finding that deleting genes for pilus and flagellum expression did not inhibit Fe(III) oxide reduction, suggests that *D. ferrophilus* has adopted strategies for extracellular electron exchange that are different from those of intensively studied electroactive microbes like *Shewanella* and *Geobacter* species. Thus, the ability to genetically manipulate *D. ferrophilus* is likely to lead to new mechanistic concepts in electromicrobiology.

**KEYWORDS** electromicrobiology, *c*-type cytochromes, e-pili, iron reduction, extracellular electron transfer, electron transfer

Address correspondence to Derek R. Lovley, dlovley@umass.edu.

*Present address: Toshiyuki Ueki, Electrobiomaterials Institute, Key Laboratory for Anisotropy and Texture of Materials (Ministry of Education), Northeastern University, Shenyang, China.

The authors declare no conflict of interest.

The mechanisms of *Desulfovibrio ferrophilus* for extracellular electron exchange are of interest because it is one of the few sulfate-reducing microorganisms available in pure culture that is also capable of conserving energy with electron transport to Fe(III) oxides (1). Fe(III) and sulfate reduction are geochemically significant processes in many anaerobic soils and sediments. Therefore, it is important to understand how microorganisms that live at redox interfaces in which the availability of Fe(III) and sulfate rapidly fluctuates may have

**FIG 1** Gene clusters in *D. ferrophilus* containing genes for putative *c*-type cytochromes that may be associated with the outer membrane. Numbers are gene numbers in the *D. ferrophilus* genome (15). Genes for *c*-type cytochromes are shown in red. DFE_0451 is an NHL repeat unit of beta-propeller protein. DFE_0463 is an uncharacterized NHL repeat domain. The localization of cytochromes predicted by PSORT (https://www.psort.org/psortb/) is shown below the genes. P and E represent predicted periplasmic and extracellular localization, respectively. Asterisks indicate that these cytochromes predicted to be localized in the periplasm were recovered in the outer membrane fraction (15).

evolved mechanisms to reduce Fe(III) as well as sulfate. Intensively studied Fe(III)-reducing microorganisms, such as *Shewanella* and *Geobacter* species, are not capable of sulfate reduction, and intensively investigated sulfate-reducing microorganisms, such as *Desulfovibrio vulgaris*, are not capable of conserving energy from Fe(III)-based respiration.

Gram-negative bacteria can reduce Fe(III) oxides either through direct electrical contacts between cell components and Fe(III) oxide or via soluble, redox-active molecules, known as electron shuttles, that accept electrons from the cell, diffuse to the Fe(III) oxide, and then transfer electrons to the Fe(III) oxide, regenerating the oxidized form of the shuttle (2). Even cells employing electron shuttles may reduce some Fe(III) oxide at the outer cell surface. However, additional mechanisms are required to overcome constraints associated with the need for outer surface electron transfer components to make electrical connections with an insoluble mineral. For example, *Shewanella oneidensis* primarily transfers electrons to Fe(III) oxide by releasing soluble extracellular flavin electron shuttles, which alleviate the need for direct contact with Fe(III) oxides (3). A similar reliance on electron shuttling was noted in *Shewanella alga* (4), *Geothrix fermentans* (5), and *Geobacter uraniireducens* (6). However, other *Geobacter* species, such as *G. metallireducens* and *G. sulfurreducens*, do not reduce Fe(III) oxide via electron shuttles (7, 8). As an alternative, they express electrically conductive pili (e-pili) that form an electrical network to expand the possibilities for electrical contact with Fe(III) oxide and extend the electronic reach of the cell (2). Although *G. sulfurreducens* also expresses cytochrome-based filaments, studies with strains constructed to express poorly conductive pili while continuing to express cytochrome-based filaments have demonstrated that it is the e-pili that are essential for Fe(III) oxide reduction (9–12).

In accordance with an electron shuttle model, *D. ferrophilus* reduces Fe(III) oxides occluded within microporous beads that prevent direct contact with the Fe(III) oxides (1). However, unlike other microbes that rely on electron shuttles for Fe(III) oxide reduction, *D. ferrophilus* also produced electrically conductive filaments with a morphology consistent with pili (1). A putative type IV pilin gene was more highly expressed during growth on Fe(III) oxide than during growth on soluble electron acceptors, a response similar to the increased pilin gene expression during growth on Fe(III) oxide by *G. metallireducens* and *G. sulfurreducens* (13, 14). These results suggested that, despite its reduction of Fe(III) oxide via an electron shuttle, the possibility that *D. ferrophilus* might also transfer electrons to Fe(III) oxides through e-pili should be explored.

In the Fe(III) oxide reduction models for Gram-negative bacteria, multiheme *c*-type cytochromes are required to transport electrons to electron shuttles or e-pili (2). Both *Shewanella* and *Geobacter* species transport electrons across the outer membrane through porin-cytochrome conduits comprised of a periplasm-facing, multiheme *c*-type cytochrome, an outer surface-facing multiheme *c*-type cytochrome, and a porin protein which couples and stabilizes the conduit. Additional *c*-type cytochromes are positioned on the outer surface to provide additional possibilities for electrical contact with extracellular electron acceptors. *D. ferrophilus* has genes for 26 multiheme *c*-type cytochromes, but none of these are homologous to those found in *Shewanella* and *Geobacter* species, and genes for porin-cytochrome conduits are not apparent (1, 15). A subset of the *D. ferrophilus* *c*-type cytochromes (Fig. 1) was proposed to be associated with the outer membrane (15), but there is substantial uncertainty about the fine-scale localization of these cytochromes. The two cytochromes predicted to be localized on the outer cell surface, DFE_0450 and DFE_0464, were not recovered in the outer membrane fraction (15). The two cytochromes identified in the outer membrane

fraction, DFE_0449 and DFE_0461, were predicted to be localized in the periplasm (15). Other multiheme *c*-type cytochromes proposed to be involved in electron exchange across the outer membrane (DFE_0448, DFE_0462, and DFE_0456) were predicted to be periplasmic (15).

For *Geobacter* species an early indication of the role of some outer surface multiheme *c*-type cytochromes in extracellular electron transfer was increased transcript abundance for the cytochrome genes during growth on Fe(III) oxide versus soluble electron acceptors (14, 16). However, transcript abundance for putative *D. ferrophilus* outer surface cytochrome genes was not higher during growth on Fe(III) oxide than during growth on soluble electron acceptors, raising doubt whether any of these cytochromes had a major role in Fe(III) oxide reduction (1).

Tools for genetic manipulation of microbes have proven invaluable for the elucidation of mechanisms for extracellular electron exchange (2). Here, we report on the first demonstration of construction of genetically modified strains of *D. ferrophilus* and provide further insights into its mechanisms for extracellular electron exchange derived from these strains.

## RESULTS AND DISCUSSION

**Elucidating likely filament composition with gene deletions.** We found that genes in the genome of *D. ferrophilus* could be replaced with a kanamycin resistance gene via double-crossover homologous recombination, as described in Materials and Methods. This methodological advance enabled evaluation of the composition and function of proteins with possible key roles in extracellular electron exchange.

For example, a previous study provided images that suggested that *D. ferrophilus* expressed polar flagella (1). These filaments as well as two types of pilus-like filaments (1) were thinner than the 30- to 50-nm-diameter membrane blebs that extend from the surface of *D. ferrophilus* grown under stress conditions (15). In our studies the putative flagella were evident with transmission electron microscopy of wild-type cells grown with sulfate as the electron acceptor (Fig. 2a and c), and the cells were motile (Fig. 2b). Analysis of the *D. ferrophilus* genome identified seven putative *fliC* genes (DFE_0352, DFE_0942, DFE_0943, DFE_1910, DFE_2105, DFE_2171, and DFE_2355) that might encode flagellin, the structural protein of flagella (see Fig. S1 and S2 in the supplemental material). Multiple *fliC* genes are also present in *D. vulgaris* Hildenborough and *Desulfovibrio desulfuricans* G20 (Fig. S2). Preventing flagellar expression by specifically eliminating all of the *D. ferrophilus* FliC genes would require identifying more antibiotic selection markers. No additional markers have been evaluated yet. However, the sigma factor FliA regulates the transcription of flagellar biosynthesis genes, including *fliC*, in other bacteria (17–19). Putative FliA-dependent promoter elements were identified for three (DFE_0352, DFE_1910, and DFE_2105) of the *D. ferrophilus* FliC genes as well as other genes encoding proteins (*ycgR*, flagellar protein YcgR; *cheW*, chemotaxis protein W; *mcp*, methyl-accepting chemotaxis protein; and genes adjacent to genes for putative flagellar assembly) associated with flagellar expression and motility (Fig. S4).

Therefore, the *D. ferrophilus* FliA gene (DFE_1249) was deleted (Fig. S3). The *fliA*-deficient mutant (Δ*fliA* strain) appeared to express shorter flagella than those in the wild-type strain, and many of the flagella were dissociated from the Δ*fliA* strain cells (Fig. 2d and e). The finding that deleting *fliA* did not completely eliminate the expression of apparent flagellar fragments is consistent with the finding that some of the FliC genes did not appear to be controlled by FliA, but rather by the sigma factor RpoN or as-yet-unknown regulators (Fig. S4). The apparent incomplete flagellar assembly in the Δ*fliA* strain was associated with the loss of motility (Fig. 2b). These results suggest that the largest-diameter filaments are flagella that are required for motility.

Two types of pilus-like filaments were apparent in previous images of *D. ferrophilus* (1), and these same filaments were again apparent in wild-type cells (Fig. 2a). One of the pilus-like filaments was relatively short and straight whereas the other type was longer and appeared to be more flexible. The short, straight filaments were previously found to be conductive, with a conductance ($0.95 \pm 0.07$ nS) lower than the conductance of *G. sulfurreducens* e-pili measured under similar conditions ($4.5 \pm 0.3$ nS) but possibly high enough to potentially play a role in long-range electron transport (1). The conductance of the longer, flexible filaments has yet to be evaluated.

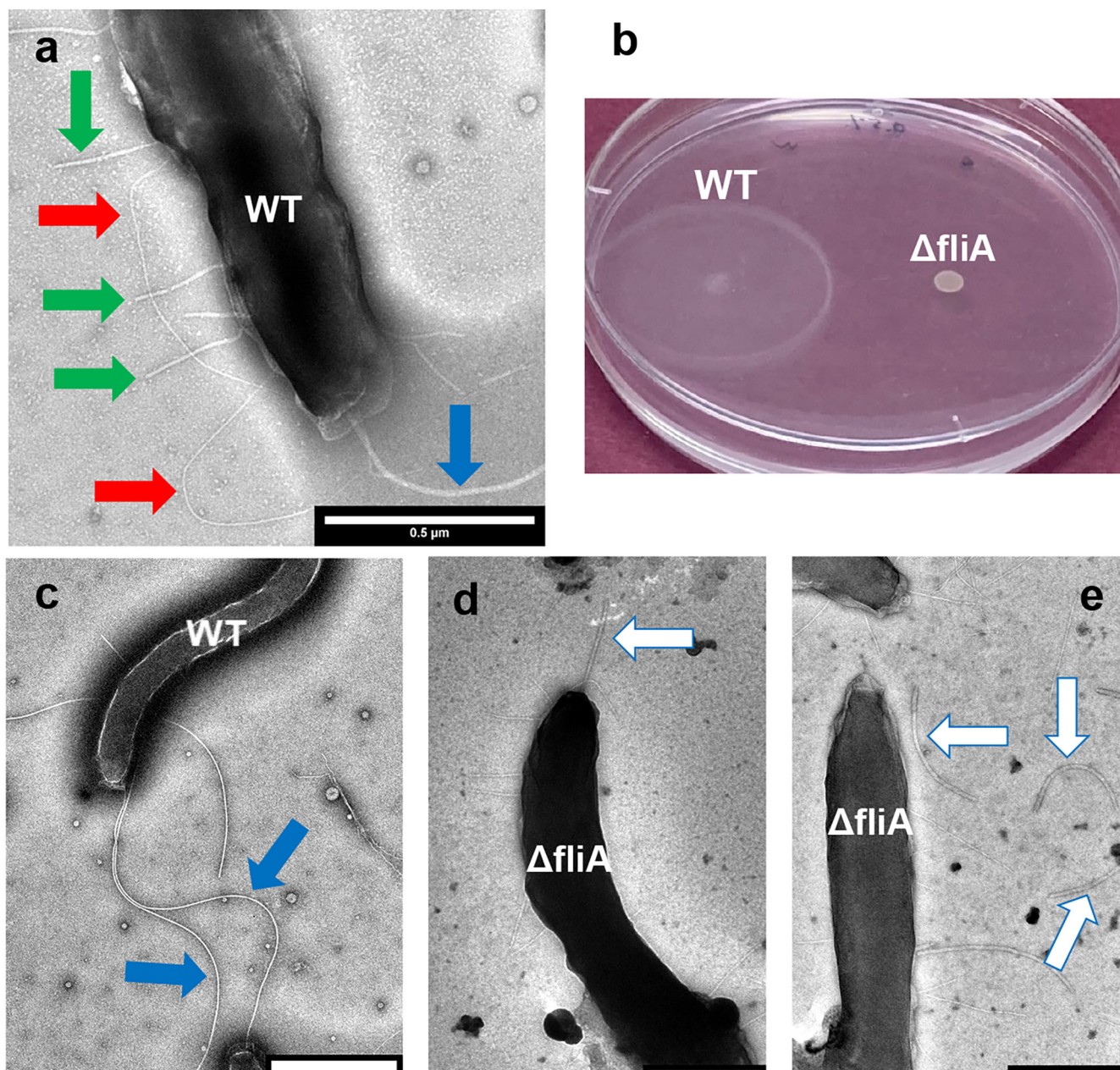

**FIG 2** *D. ferrophilus* filament expression and motility. (a and c) Transmission electron micrographs of a typical wild-type (WT) cell. Colored arrows highlight features as follows: blue arrow, putative flagella; red arrow, long flexible putative pili; green arrow, short, straight putative pili. (b) Motility assay of wild-type (WT) and Δ*fliA* strains. One of duplicate plates (0.5% agar) is shown. Similar results were observed with an 0.7% agar concentration (data not shown). (d and e) Transmission electron micrographs of Δ*fliA* strain cells. White arrows highlight truncated flagella, some of which are disassociated from the cells.

In order to obtain information on the possible composition of the two pilus-like filaments, the *D. ferrophilus* genome was searched for putative pilin genes. Genes for possible type IV pilins were identified (Fig. 3). The genes DFE_1797 to DFE_1801 are located next to other genes for type IV pilus assembly and are predicted to be type IV pilins. Transcript abundance for the putative pilin gene DFE_1797 was ca. 4- to 6-fold higher in cells grown with Fe(III) oxide than in cells grown with sulfate or Fe(III) citrate as the electron acceptor, suggesting a possible role in Fe(III) oxide reduction (1).

A strain in which the genes DFE_1797 to DFE_1801 were deleted, designated the Δpil5 strain, continued to express flagella and the shorter, straight filaments but not the long flexible filaments (Fig. 4a). This requirement for putative pilin genes for *D. ferrophilus* to express these

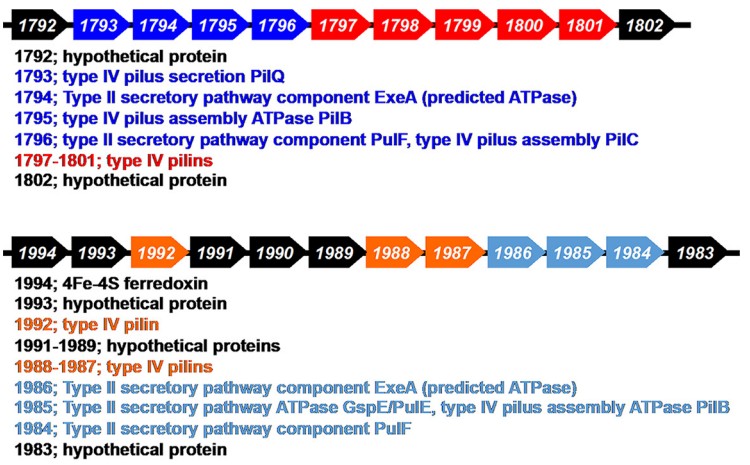

**FIG 3** Gene clusters containing putative *D. ferrophilus* pilin genes.

filaments suggested that the long, flexible filaments are type IV pili and that the shorter, straighter filaments, which were previously found to be conductive, were comprised of some other protein.

We designated genes DFE_1987, DFE_1988, and DFE_1992 as pseudopilins because unlike the DFE_1797 to DFE_1801 cluster, these genes were adjacent to genes that might have a role in a type II secretion system (Fig. 3). Strains lacking genes DFE_1987 and DFE_1988 (Δspp) or DFE_1992 (Δlpp) expressed the long flexible filaments but not the straight shorter filaments (Fig. 4b and c). These results suggest that the putative pseudopilins are required for the expression of the short, straight filaments that were previously (1) found to be conductive. Pseudopili are typically involved in the secretion of exoproteins (20, 21). Therefore, at present it is not possible to conclude whether the short straight filaments are comprised of pseudopilins or of other proteins that require pseudopili for extrusion outside the cell. For simplicity these filaments are referred to as pili in subsequent discussion.

**Eliminating pili, cytochromes, or flagellum-based motility does not inhibit Fe(III) oxide reduction.** Wild-type *D. ferrophilus* readily reduced Fe(III) with lactate as the electron donor but not in its absence (Fig. 5; Fig. S5). The phenotypes of various gene deletion mutants were consistent with the previous finding (1) that *D. ferrophilus* can reduce Fe(III) oxide via an electron shuttle, a mode of Fe(III) oxide reduction that does not require flagellum-based motility or e-pili. For example, strains that lacked the thin long putative type IV pili (Δpil5 strain), or the shorter, straighter pili (Δspp and Δlpp strains) reduced Fe(III) oxide as well as the wild type did (Fig. 5a). This result demonstrates that neither of the

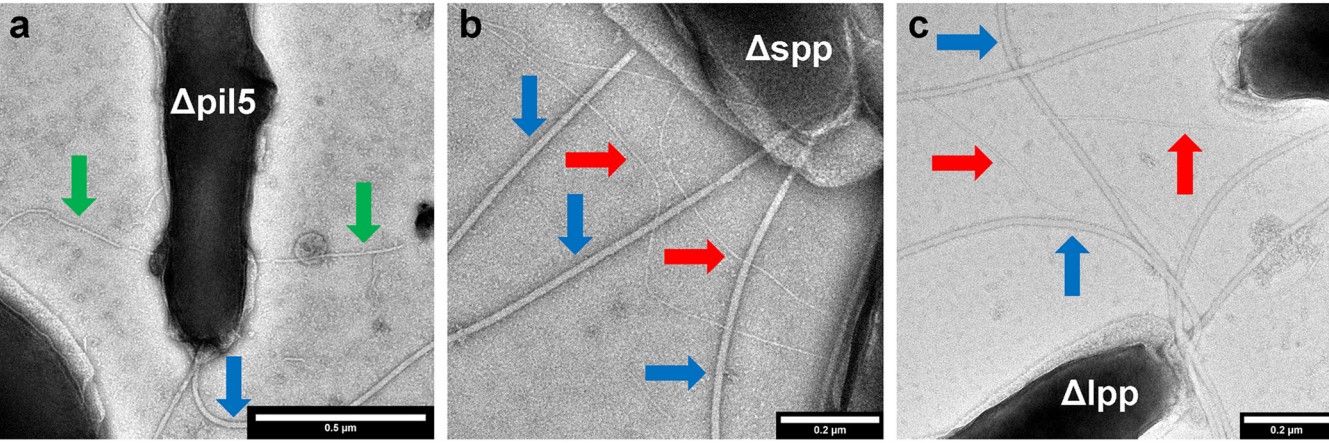

**FIG 4** Impact of gene deletions on *D. ferrophilus* pilus expression. Transmission electron micrographs of Δpil5 strain (a), Δspp strain (b), and Δlpp strain (c). Colored arrows highlight features as follows: blue arrow, putative flagella; red arrow, long flexible putative pili; green arrow, short, straight putative pili.

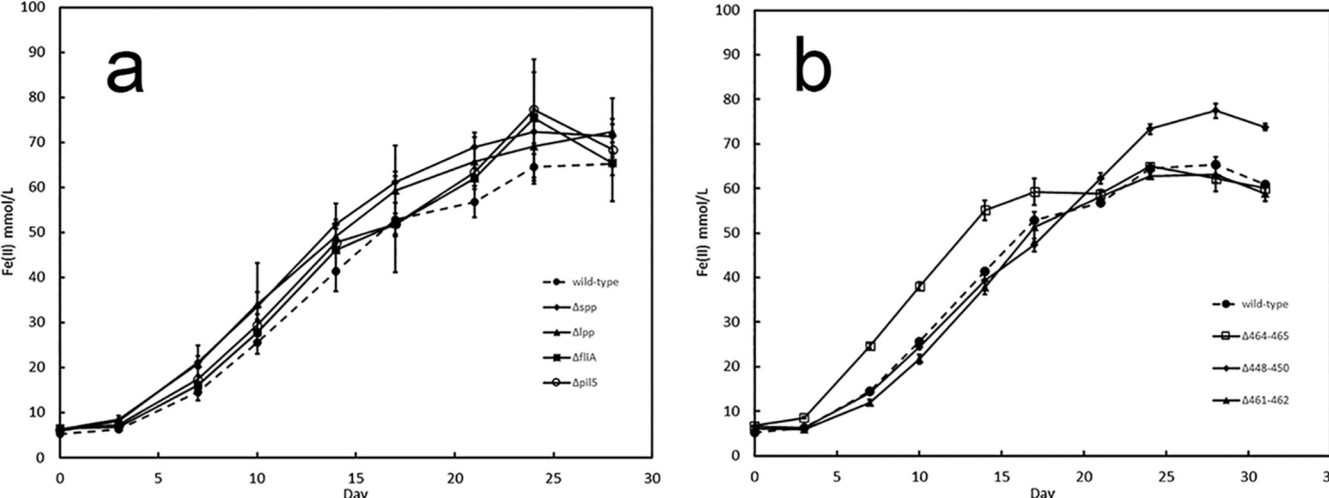

**FIG 5** Fe(III) oxide reduction by wild-type and mutant strains of *D. ferrophilus*. (a) Fe(II) produced by the wild-type strain and mutant strains defective in the expression of pili or flagella. (b) Fe(II) produced by the wild-type strain and mutant strains in which the designated cytochrome genes were deleted. Results are the mean and standard deviations from triplicate cultures for each strain.

two types of pili emanating from *D. ferrophilus* is essential for Fe(III) oxide reduction. It is possible that both filament types might contribute to long-range electron transport and that eliminating the possibility for expression of just one of the filaments is not sufficient to inhibit Fe(III) oxide reduction because the other pilus type is still available. Evaluation of this possibility will require the development of genetic tools for making multiple gene deletions in the same *D. ferrophilus* strain. However, it is expected that a microbe like *D. ferrophilus* that produces an electron shuttle would not also require e-pili for Fe(III) oxide reduction. For example, *G. uraniireducens*, which effectively reduces Fe(III) oxide by producing an electron shuttle, does not express e-pili (6), in contrast to its close relatives *G. metallireducens* and *G. sulfurreducens*, which require e-pili (9, 10, 22–24) because they lack electron shuttles (7, 8).

Flagellum-based motility enhances Fe(III) oxide reduction by *G. metallireducens* and *G. sulfurreducens*, possibly because it enables cells to search for and establish contact with Fe(III) oxides (13, 25, 26). To determine the importance of flagellum-based motility for *D. ferrophilus* Fe(III) oxide reduction, the ability of the nonmotile Δ*fliA* strain to reduce Fe(III) oxide was evaluated (Fig. 5a). The nonmotile strain reduced Fe(III) oxide as well as the wild type did. This phenotype is also consistent with an electron shuttle mechanism for Fe(III) oxide reduction because electron shuttles eliminate the necessity for motility to intimately colocalize cells and Fe(III) oxides.

Outer surface, multiheme, *c*-type cytochromes are typically essential electron transport components for Fe(III) oxide reduction by Gram-negative bacteria whether they rely on electron shuttles or on e-pili to extend their range of electron transport beyond the immediate cell surface (2, 27, 28). To determine whether any of the previously recognized (15) multiheme cytochromes thought to be localized near the cell outer surface (Fig. 1) might be involved in Fe(III) oxide reduction, three strains were constructed: strain Δ448-450 (deletion of DFE_0448 to DFE_0450), strain Δ461-462 (deletion of DFE_0461 and DFE_0462), and strain Δ464-465 (deletion of DFE_0464 and DFE_0465). None of these strains was defective in Fe(III) oxide reduction (Fig. 5b), indicating that none of the putative outer surface cytochromes is essential for Fe(III) oxide reduction. One possibility for the continued Fe(III) oxide reduction in the cytochrome deletion mutants is that cytochromes with genes in different clusters might compensate for the loss of deleted cytochromes. However, it was also previously found that all six of the putative outer surface cytochrome genes whose transcript abundance was quantified (DFE_0465 transcripts were not evaluated) did not have higher gene transcript abundance during growth on Fe(III) oxide than during growth on sulfate (1). Outer surface cytochromes are not required for sulfate reduction. Thus, it would be expected that a cell might specifically regulate the production of biosynthetically

expensive cytochromes to minimize their expression during sulfate reduction and increase their production during Fe(III) oxide reduction if the cytochromes were essential to reduce Fe(III) oxides. Alternative strategies that do not require outer surface cytochromes for the reduction of electron shuttles are possible, as exemplified by Gram-positive bacterial reduction of electron shuttles with flavoproteins (2, 29). Therefore, the lack of cytochrome gene deletion impact on Fe(III) oxide reduction may further reflect reliance of *D. ferrophilus* on electron shuttling as its primary mechanism for Fe(III) oxide reduction.

*D. ferrophilus* reduction of the electron shuttle within the cell, rather than with putative outer surface cytochromes, would also be consistent with the observation that, following the addition of an exogenous electron shuttle, *D. ferrophilus* reduces Fe(III) oxide occluded within porous beads (1) much more slowly than *Geobacter* species (6–8), which are thought to reduce electron shuttles at the outer cell surface. A requirement for an electron shuttle to move across the outer membrane would make electron shuttling slower than if the electron shuttle was reduced with cytochromes at the outer cell surface.

**Implications.** Few electroactive microbes are currently genetically tractable, limiting the exploration of the diversity of mechanisms for extracellular electron exchange (2). Thus, the demonstration of a simple strategy for making gene deletions in *D. ferrophilus* provides an opportunity to evaluate mechanisms for electroactivity in a microbe that has substantial differences from intensively studied electroactive microbes like *Shewanella* and *Geobacter* species.

The phenotypes of mutations that prevented flagellum-based motility, as well as pilus and cytochrome expression, are consistent with the concept that *D. ferrophilus* primarily relies on a soluble electron shuttle for electron transfer to Fe(III) oxides. Further development of genetic tools to permit deleting multiple regions of the chromosome in the same strain is required to determine whether functional redundancy of outer surface cytochromes or the two different types of pili might explain the finding that no individual cytochrome or pilus type evaluated was essential for Fe(III) oxide reduction. However, it has already been demonstrated that *D. ferrophilus* can reduce Fe(III) oxides that it cannot directly contact (1), and thus, electrical contacts via pili or outer surface cytochromes may not be necessary. The results emphasize that the presence of genes for multiheme *c*-type cytochromes cannot be simply interpreted as evidence for involvement of *c*-type cytochromes in extracellular electron exchange. They also can function in intermediary intracellular electron transfer (30), intracellular reduction of metal ions (31), and temporary intracellular electron storage to permit continued respiration when electron acceptors are unavailable (32).

In addition to its ability to grow with Fe(III) oxide as an electron acceptor, another unique feature of *D. ferrophilus* (previously known as strain IS5) is its ability to rapidly corrode metallic iron (33). This was originally attributed to direct electron uptake from Fe(0) (33). However, direct electron uptake was only inferred (34, 35). Subsequent studies concluded that *D. ferrophilus* did not directly accept electrons from Fe(0) (1). Instead, its electron donor was the $H_2$ that is generated from Fe(0) and serves as an electron shuttle between Fe(0) and the cells. However, it has been suggested that *D. ferrophilus* can directly accept electrons from negatively poised electrodes without $H_2$ serving as an intermediary electron carrier (36–38). Expansion of the genetic approach described here should make it feasible to more rigorously address these additional questions about *D. ferrophilus* extracellular electron exchange by generating the appropriate mutant strains.

## MATERIALS AND METHODS

**Strains and growth conditions.** *D. ferrophilus* IS5 was obtained from DSMZ-German Collection of Microorganisms and Cell Cultures GmbH. It served as the parent (wild-type) strain for the construction of mutants. *D. ferrophilus* was anaerobically grown at 30°C in DSMZ 195c medium with modifications as previously described (1). Sodium DL-lactate (60 mM) served as the electron donor with either sodium sulfate (30 mM) or poorly crystalline Fe(III) oxide (100 mmol/L) as the electron acceptor. The poorly crystalline Fe(III) oxide was prepared as previously described (39) by slowly neutralizing a solution of ferric chloride and collecting and washing the resulting precipitate. This form of Fe(III) oxide is highly available for microbial reduction and is typically employed when evaluating the ability of isolates to reduce Fe(III) oxides (40). The gas phase was $N_2/CO_2$ (80:20). Cysteine (1 mM) was added as a reductant when sulfate was the electron acceptor.

**Construction of *D. ferrophilus* mutant strains.** To construct the Δ*fliA* strain, ~1-kb upstream and downstream regions of the *fliA* gene were amplified by PCR with primer pairs fliA-up1/fliA-up2 and fliA-do1/

fliA-do2 (see Table S1 in the supplemental material), respectively, with the genomic DNA of *D. ferrophilus* IS5 as the template. A kanamycin resistance gene was amplified by PCR with a primer pair, km-F/km-R (Table S1), and the plasmid pBBR1MCS-2 (41) as the template. The PCR products were digested with restriction enzymes, ligated with T4 DNA ligase, and cloned in the plasmid pBluescript (Stratagene). Correct sequences of the cloned PCR products were confirmed by DNA sequencing. The plasmid containing the upstream and downstream regions of the *fliA* gene and the kanamycin resistance gene was linearized by XbaI. The *fliA* gene was replaced with the kanamycin resistance gene via double-crossover homologous recombination (Fig. S3).

Transformation of *D. ferrophilus* was carried out by electroporation. Electrocompetent *D. ferrophilus* cells were prepared from cultures grown in the lactate-sulfate medium. All manipulations were conducted in an anaerobic glove bag (7% $H_2$/20% $CO_2$/73% $N_2$). The cultures (10 mL) at the mid-log phase (~0.3 value of optical density at 600 nm) were harvested in a centrifuge tube with an O-ring screw cap by centrifugation at 4°C. The cells were washed with a buffer (1 mM $MgCl_2$/175 mM sucrose/1 mM HEPES, pH 7) twice. The washed cells were resuspended in 50 $\mu$L and kept on ice. The linearized plasmid (1 $\mu$g) was mixed with the washed cells, and the mixture was electroporated in an electroporation cuvette (0.1-cm gap) at 1.5 kV by a MicroPulser electroporator (Bio-Rad). The cells were recovered in the lactate-sulfate medium and incubated at 30°C for 24 h. Transformants were selected with the lactate-sulfate medium containing 1.5% agar supplemented with Geneticin (G418) (400 $\mu$g/mL). Typically, 30 to 60 colonies per plate were recovered. Proper recombination was confirmed by PCR with primer pairs fliA-P1/fliA-P2, fliA-P1/fliA-P3, and fliA-do2/km-V (P4 and P5 in Fig. S3).

Other mutants were constructed as described above. Primers used for the strain constructions are shown in Table S1.

**Growth measurement, motility assay, and microscopy.** Growth of *D. ferrophilus* with sulfate as the electron acceptor was monitored by measuring the optical density at 600 nm. Reduction of Fe(III) was monitored by measuring the concentration of Fe(II) by the ferrozine assay as described previously (39). The motility assay was conducted on the lactate-sulfate medium containing 0.5% or 0.7% agar at 30°C in the anaerobic glove bag. The cultures (5 $\mu$L) at the mid-log phase were spotted on the agar surface. Photographs were taken after 4 days (0.5% agar) or 9 days (0.7% agar) of incubation. Transmission electron microscopy was performed as described previously (1).

**Bioinformatic tools.** Sequence similarity was analyzed by NCBI blastp (protein-protein BLAST) (https://blast.ncbi.nlm.nih.gov). Genome sequences were utilized from NCBI for *D. ferrophilus* IS5 (NCBI reference sequence NZ_AP017378.1), *D. vulgaris* Hildenborough (NCBI reference sequence NC_002937.3), and *D. desulfuricans* G20 (NCBI reference sequence NC_007519.1). Sequence alignments were constructed by Clustal Omega Multiple Sequence Alignment (https://www.ebi.ac.uk/Tools/msa/clustalo/). Protein localization was predicted by PSORTb (https://www.psort.org/psortb/).

## SUPPLEMENTAL MATERIAL

Supplemental material is available online only.

**SUPPLEMENTAL FILE 1**, PDF file, 0.5 MB.

## ACKNOWLEDGMENTS

Construction of pilin-gene deletion mutants was supported by National Science Foundation grant DMR-2027102.

The transmission electron microscopy images were collected in the Electron Microscopy Facility at the Institute for Applied Life Sciences, UMass Amherst.

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
