## [Reviewer comments · Microbiology Spectrum]

Microbiology Spectrum

Genetic Manipulation of *Desulfovibrio ferrophilus* and Evaluation of Fe(III) Oxide Reduction Mechanisms

Toshiyuki Ueki, Trevor Woodard, and Derek Lovley

Corresponding Author(s): Derek Lovley, University of Massachusetts Amherst

Review Timeline:

Submission Date:	September 26, 2022
Editorial Decision:	November 1, 2022
Revision Received:	November 9, 2022
Accepted:	November 11, 2022

Editor: Daniel Bond

Reviewer(s): The reviewers have opted to remain anonymous.

Transaction Report:

DOI: <https://doi.org/10.1128/spectrum.03922-22>

November 1, 2022

Dr. Derek R Lovley
University of Massachusetts Amherst
Microbiology
Morrill Science Center IVN
Amherst, MA 01003

Re: Spectrum03922-22 (Genetic Manipulation of *Desulfovibrio ferrophilus* and Evaluation of Fe(III) Oxide Reduction Mechanisms)

Dear Dr. Derek R Lovley:

Thank you for submitting your manuscript to Microbiology Spectrum. This manuscript has been reviewed by two external reviewers who both agreed these methods and initial results with this new system are a valuable contribution, and had no significant issues. I am requesting modifications so you may answer minor questions from Reviewer #1 about methodology by adding appropriate text to the manuscript, and, if available, provide a statistical comparison. Reviewer #2 suggested additional discussion points but had no substantial changes.

Spectrum policy asks that when submitting the revised version of your paper, please provide (1) point-by-point responses to the issues raised by the reviewers as file type "Response to Reviewers," not in your cover letter, and (2) a PDF file that indicates the changes from the original submission (by highlighting or underlining the changes) as file type "Marked Up Manuscript - For Review Only". Please use this link to submit your revised manuscript - we strongly recommend that you submit your paper within the next 60 days or reach out to me. Detailed instructions on submitting your revised paper are below.

Link Not Available

Sincerely,

Daniel Bond

Journals Department
Reviewer comments:

Reviewer #1 (Comments for the Author):

I reviewed 'Genetic Manipulation of *Desulfovibrio ferrophilus* and Evaluation of Fe(III) Oxide Reduction Mechanisms' by Ueki, Woodard, and Lovley. The manuscript describes the application of electroporation-mediated transformation methods to generate gene deletions in *D. ferrophilus* via homologous recombination. Using this technique, Ueki and colleagues disrupted flagella, pili, pseudopili, and cytochrome expression and tested these mutants for loss of capacity for Fe(III) oxide reduction to begin to

elucidate the mechanism(s) of EET in this organism.

Overall, the manuscript is clearly and concisely written. The authors discuss the importance of having genetically tractable pure culture models for Fe(III) oxide reduction, but provide little information on the development or optimization of establishing this organism as 'genetically tractable.'

With regard to EET mechanisms, the experiments conducted and presented in this manuscript provide a first pass at attempting to prove that this organism engages in EET via soluble electron carriers and not via OMCs or conductive pili.

General comments:

- Please provide information on the efficiency of electroporation and any other optimizations undertaken to create a genetically tractable system in *D. ferrophilus*.
- Please add if you attempted to complement the gene deletions using a different selection marker, or has complementation not yet been optimized?
- Add if you tested the mutants you generated in bioelectrochemical systems?
- Is there any information showing if they possess the same capacity for EET to an anode as they do in culture with iron(III) oxide?

Specific comments:

Line 182-184: You state that only one antibiotic selection marker has been identified. Can you add if other antibiotic selection markers been tested, indicating if it was possible to create a double knockout mutant with two different antibiotic resistance genes? Please provide more detail.

Line 202-205: Is there any evidence that the longer, more flexible filaments were tested for conductivity?

Line 264-268: Yes, none of the deletion strains were defective in Fe(III) oxide reduction, but Figure 5b shows that wild type produces nearly 15 mM more Fe(II) compared to all of the cytochrome deletion mutants. This suggests that deletion of just one cytochrome is potentially deleterious to the cells. Does this suggest that overproduction of other cytochromes can not compensate for the loss of deleted cytochromes?

Line 265-266: What is meant by "long-range extracellular electron transfer" in this context? This experiment tested whether cytochrome mutants could reduce poorly crystalline Fe(III) oxide in the media, not how far electrons could be transferred out of the cell.

Figure 5: Please perform statistical analyses to test whether iron(III) oxide reduction capacity is affected by deletion of pili/flagella (5a) or various cytochrome genes (5b) and incorporate into the text as appropriate.

Figure 5b: $\Delta 464-465$ Add a comment or explanation for why strains have decreased lag time compared to other strains, including wild type.

Reviewer #2 (Comments for the Author):

The manuscript by Ueki et al describes the construction of *Desulfovibrio ferrophilus* mutants deprived of partial flagella movement, pili-like proteins and some periplasmic or outer-membrane located multi-heme c-type cytochromes (MHCs). The wild type was previously known to not only reduce sulfate but also iron oxide. The results showed that deletion of any of these individual components did not influence the capacity of iron oxide reduction and it was speculated that in combination with previous results obtained in the same lab the redox shuttles mediate the extracellular electron transfer for Fe(III) reduction. The 'main achievement' is the construction of mutants by seemingly a simple approach, replacement of the targeted genes by antibiotic resistance gene-containing plasmid fragment through electroporation. Production of these mutants and the tests on Fe(III) reduction are very interesting and provide a new insight into the flexibility of extracellular electron transfer routes in different organisms.

My first comment is the lack of details about the preparation of the so called poorly crystalline Fe(III) oxide. The physicochemical properties of iron oxide are not only important in terms of redox potential but also influence the accessibility to microbial cells and hence should be described in detail.

My second comment is about the possible function of MHCs. The functions of flagella and pili can be imagined even if they are not associated with the extracellular electron transfer. But what can be the function of those periplasmic and outer membrane-localized MHCs if they are not for electron transfer. Are there other phenotypes that were not tested, such as could they be involved in sulfate reduction? A few discussions about the variety of strategies shall be welcome by audience.

Staff Comments:

Preparing Revision Guidelines

Please return the manuscript within 60 days; if you cannot complete the modification within this time period, please contact me. If you do not wish to modify the manuscript and prefer to submit it to another journal, please notify me of your decision immediately so that the manuscript may be formally withdrawn from consideration by Microbiology Spectrum.

Response to Reviewer Comments (Reviewer comments in bold)

Reviewer #1.

- **Please provide information on the efficiency of electroporation and any other optimizations undertaken to create a genetically tractable system in *D. ferrophilus*.**

We were fortunate that the first set of electroporation conditions that were evaluated was successful. It is now noted in the revised manuscript that this procedure yielded 30-60 transformant colonies per plate. Attempts for further optimization did not seem necessary.

- **Please add if you attempted to complement the gene deletions using a different selection marker, or has complementation not yet been optimized?**

We attempted to introduce a plasmid into *D. ferrophilus* with a modification of the plasmid pMO9075, which is known to be maintained in some *Desulfovibrio* species (Keller KL, Rapp-Giles BJ, Semkiw ES, Porat, I, Brow SD, Wall, JD. 2014. New model for electron flow for sulfate reduction in *Desulfovibrio alaskensis* G20. 80:855-868.), but this was unsuccessful, suggesting that a derivative plasmid of pMO9075 is not applicable to *D. ferrophilus* and/or the conditions for the electroporation of a plasmid need to be optimized. If we could introduce a plasmid into *D. ferrophilus*, then we would test other antibiotic resistances and attempt complementation. Unfortunately, such studies will be delayed for some time as the first author of this study has moved to a new lab.

- **Add if you tested the mutants you generated in bioelectrochemical systems?**

None of the mutants were tested in bioelectrochemical systems. Previous studies (Liang D, Liu X, Woodard TL, Holmes DE, Smith JA, Nevin KP, Feng Y, Lovley DR. 2021. Extracellular electron exchange capabilities of *Desulfovibrio ferrophilus* and *Desulfopila corrodens*. Environmental Science and Technology 55:16195–16203) have demonstrated that *D. ferrophilus* produces relatively low current densities (ca. 100-fold lower than *G. sulfurreducens* under comparable conditions) and thus current production was not of sufficient interest to warrant investigation.

- **Is there any information showing if they possess the same capacity for EET to an anode as they do in culture with iron(III) oxide?**

Current production was not included in this study because current production by *D. ferrophilus* is not a property of significant interest at this time. *D. ferrophilus* produces low current densities and is not closely related to microbes that are typically enriched on current-harvesting anodes.

Specific comments:

Line182-184: You state that only one antibiotic selection marker has been identified. Can you add if other antibiotic selection markers been tested, indicating if it was possible to create a double knockout mutant with two different antibiotic resistance genes? Please provide more detail.

As suggested, the text has been modified to indicate that no additional antibiotic selection markers have been evaluated yet. Unfortunately, such studies will be delayed for some time as the first author of this study has moved to a new lab.

Line 202-205: Is there any evidence that the longer, more flexible filaments were tested for conductivity?

The conductivity of the longer, more flexible filaments has not been evaluated. The text has been modified to make this clearer.

Line 264-268: Yes, none of the deletion strains were defective in Fe(III) oxide reduction, but Figure 5b shows that wild type produces nearly 15 mM more Fe(II) compared to all of the cytochrome deletion mutants. This suggests that deletion of just one cytochrome is potentially deleterious to the cells. Does this suggest that overproduction of other cytochromes can not compensate for the loss of deleted cytochromes?

In contrast to the reviewer's comment, the wild-type strain does not produce more Fe(II) than the mutants. The wild-type strain (designated with solid circles connected with dashed line) produces the same amount of Fe(II) as two of the mutants. Therefore, none of the cytochrome genes that were deleted appear to be required for Fe(III) oxide reduction as effective as the wild-type strain.

One of the mutants ($\Delta 448-450$) does produce more Fe(II) than the wild-type and the two other mutants near the end of the incubation. However, the rate of Fe(II) accumulation with the $\Delta 448-450$ mutant matches the rate of Fe(II) accumulation of the wild-type for most of the incubation. Another mutant ($\Delta 464-465$) has a shorter lag than the other strains. Neither of these phenotypes negate the conclusion that none of the cytochromes appear to be essential for Fe(III) oxide reduction.

Line 265-266: What is meant by "long-range extracellular electron transfer" in this context? This experiment tested whether cytochrome mutants could reduce poorly crystalline Fe(III) oxide in the media, not how far electrons could be transferred out of the cell.

The text has been modified to replace "long-range extracellular electron transfer" with the phrase "Fe(III) oxide reduction."

Figure 5: Please perform statistical analyses to test whether iron(III) oxide reduction capacity is affected by deletion of pili/flagella (5a) or various cytochrome genes (5b) and incorporate into the text as appropriate.

There is no instance in Figure 5 in which any of the mutants produced less Fe(II) than the wild-type strain. Therefore, in no instance can it be concluded that any of the genes deleted are essential for Fe(III) oxide reduction. The results as presented provide the mean and standard deviation for triplicate incubations. No other statistical analysis is warranted.

Figure 5b: $\Delta 464-465$ Add a comment or explanation for why strains have decreased lag time compared to other strains, including wild type.

Small differences in lag time for Fe(III) oxide reduction are not considered to be a meaningful phenotype. This is particularly the case when evaluating whether a gene is essential for the phenotype being examined. Regardless of the shorter lag time, the conclusion that those cytochrome genes are not essential for Fe(III) oxide reduction still holds.

My first comment is the lack of details about the preparation of the so called poorly crystalline Fe(III) oxide. The physicochemical properties of iron oxide are not only important in terms of redox potential but also influence the accessibility to microbial cells and hence should be described in detail.

As suggested, the method of preparation has now been described. We also now explain that it is known that this form of Fe(III) oxide is readily available for microbial reduction and is the form of Fe(III) oxide that has been routinely employed in analyses of microbial Fe(III) oxide reduction for over 30 years.

My second comment is about the possible function of MHCs. The functions of flagella and pili can be imagined even if they are not associated with the extracellular electron transfer. But what can be the function of those periplasmic and outer membrane-localized MHCs if they are not for electron transfer. Are there other phenotypes that were not tested, such as could they be involved in sulfate reduction? A few discussions about the variety of strategies shall be welcome by audience.

As suggested, a sentence has been added to state that multi-heme *c*-type cytochromes can function in intermediary intracellular electron transfer, intracellular reduction of metal ions, and temporary intracellular electron storage to permit continued respiration when electron acceptors are unavailable.

November 11, 2022

Dr. Derek R Lovley
University of Massachusetts Amherst
Microbiology
Morrill Science Center IVN
Amherst, MA 01003

Re: Spectrum03922-22R1 (Genetic Manipulation of *Desulfovibrio ferrophilus* and Evaluation of Fe(III) Oxide Reduction Mechanisms)

Dear Dr. Derek R Lovley:

I am pleased to inform you that after reviewing changes to your manuscript, it has been accepted and I am forwarding it to the ASM Journals Department for publication. You will be notified when your proofs are ready to be viewed.

Sincerely,

Daniel Bond
Editor, Microbiology Spectrum
